DOI: 10.1038/s41467-018-07433-1　　**OPEN**

# Quantum-enhanced sensing using non-classical spin states of a highly magnetic atom

Thomas Chalopin[1], Chayma Bouazza[1], Alexandre Evrard[1], Vasiliy Makhalov[1], Davide Dreon[1,2], Jean Dalibard[1], Leonid A. Sidorenkov[1,3] & Sylvain Nascimbene[1]

Coherent superposition states of a mesoscopic quantum object play a major role in our understanding of the quantum to classical boundary, as well as in quantum-enhanced metrology and computing. However, their practical realization and manipulation remains challenging, requiring a high degree of control of the system and its coupling to the environment. Here, we use dysprosium atoms—the most magnetic element in its ground state—to realize coherent superpositions between electronic spin states of opposite orientation, with a mesoscopic spin size $J = 8$. We drive coherent spin states to quantum superpositions using non-linear light-spin interactions, observing a series of collapses and revivals of quantum coherence. These states feature highly non-classical behavior, with a sensitivity to magnetic fields enhanced by a factor 13.9(1.1) compared to coherent spin states—close to the Heisenberg limit $2J = 16$—and an intrinsic fragility to environmental noise.

[1] Laboratoire Kastler Brossel, Collège de France, CNRS, ENS-PSL University, Sorbonne Université, 11 Place Marcelin Berthelot, 75005 Paris, France. [2] Present address: Department of Physics, ETH Zurich, 8093 Zurich, Switzerland. [3] Present address: SYRTE, Observatoire de Paris, PSL University, CNRS, Sorbonne Université, LNE, 61 avenue de l'Observatoire, 75014 Paris, France. Correspondence and requests for materials should be addressed to L.A.S. (email: leonid.sidorenkov@obspm.fr)

Future progress in quantum technologies is based on the engineering and manipulation of physical systems with highly non-classical behavior[1], such as quantum coherence[2], entanglement[3], and quantum-enhanced metrological sensitivity[4,5]. These properties generally come together with an inherent fragility due to decoherence via the coupling to the environment, which makes the generation of highly non-classical states challenging[6]. An archetype of such systems consists in an object prepared in a coherent superposition of two distinct quasi-classical states, realizing a conceptual instance of Schrödinger cat[7]. Such states have been realized in systems of moderate size—referred to as 'mesoscopic' hereafter—with trapped ions[8,9], cavity quantum electrodynamics (QED) systems[10–12], superconducting quantum interference devices[13], optical photons[14–17], and circuit QED systems[18,19]. Non-classical behavior can also be achieved with other types of quantum systems, including squeezed states[20–31].

Inspired by the hypothetical cat state |dead⟩+|alive⟩ introduced by Schrödinger in his famous Gedanken experiment, one usually refers to a cat state in quantum optics as a superposition of quasi-classical states consisting in coherent states of the electromagnetic field, well separated in phase space and playing the role of the |dead⟩ and |alive⟩ states[7]. Such cat states can be dynamically generated in photonic systems, e.g. using a Kerr non-linearity[18,32]. For a spin $J$, a quasi-classical coherent state is represented as a state $|\pm J\rangle_{\hat{\mathbf{u}}}$ of maximal spin projection $m = \pm J$ along an arbitrary direction $\hat{\mathbf{u}}$. It constitutes the best possible realization of a classical state of well-defined polarization, as it features isotropic fluctuations of the perpendicular spin components, of minimal variance $\Delta J_{\hat{\mathbf{v}}} = \sqrt{J/2}$ for $\hat{\mathbf{v}} \perp \hat{\mathbf{u}}$[33]. A cat state can then be achieved for large $J$ values, and it consists in the coherent superposition of two coherent spin states of opposite magnetization, which are well separated in phase space. We mention that the Hilbert space dimension of $2J + 1$ scales linearly with the separation between the two coherent states of the superposition. Such cat states can be created under the action of non-linear spin couplings[34–37]. These techniques have been implemented with individual alkali atoms, using laser fields to provide almost full control over the quantum state of their hyperfine spin[38–42]. However, the small spin size involved in these systems intrinsically limits the achievable degree of non-classical behavior.

Non-classical spin states have also been created in ensembles of one-electron and two-electron atoms[5]. When each atom carries a spin-1/2 degree of freedom, a set of $N$ atoms evolving identically can collectively behave as an effective spin $J = N/2$, that can be driven into non-classical states via the interactions between atoms[34–37,43]. In such systems, spin-squeezed states have been realized experimentally[20–22,25,26,28–31], as well as non-gaussian entangled states[44]. Yet, cat states remain out of reach due to their extreme sensitivity to perturbations in such systems. This behavior results from the large size $2^N$ of the Hilbert space (when taking into account non-symmetric quantum states), which scales exponentially with the system size $N$, resulting in a large number of decoherence channels (e.g. losing a single particle fully destroys their quantum coherence).

In this work, we use samples of dysprosium atoms, each of them carrying an electronic spin of mesoscopic size $J = 8$. We exploit the AC Stark shift produced by off-resonant light[38] to drive non-linear spin dynamics. Each atomic spin independently evolves in a Hilbert space of dimension $2J + 1 = 17$, much smaller than the dimension $2^N \sim 10^5$ of an equivalent system of $N = 16$ spins 1/2. We achieve the production of quantum superpositions of effective size 13.9(1.1) (as defined hereafter), close to the maximum allowed value $2J = 16$ for a spin $J$. As this size can be considered large, but not macroscopic according to the original

Schrödinger idea, we will hereafter refer to such quantum superpositions as Schrödinger kitten states[45]. We provide a tomographic reconstruction of the full density matrix of these states and monitor their decoherence due to the dephasing induced by magnetic field noise.

## Results

**Experimental protocol.** Our experimental scheme is sketched in Fig. 1a. We use an ultracold sample of about $10^5$ $^{164}$Dy atoms, initially spin-polarized in the absolute ground state $|-J\rangle_z$, under a quantization magnetic field $\mathbf{B} = B\hat{\mathbf{z}}$, with $B = 18.5(3)$ mG (see Methods). The non-linear spin dynamics results from spin-dependent energy shifts induced by a laser beam focused on the atomic sample. The laser wavelength is chosen close to the 626-nm resonance line, such that the light shifts are proportional to the polarizability tensor of a $J = 8$ to $J' = 9$ optical transition. For a linear light polarization along $x$, the light shift operator reduces to a coupling $\propto J_x^2$ (up to a constant), and we expect the spin dynamics to be described by the Hamiltonian[38]

$$\hat{H} = \hbar\omega_{\mathrm{L}}\hat{J}_z + \hbar\omega\hat{J}_x^2, \qquad (1)$$

where the first term corresponds to the Larmor precession induced by the magnetic field, and the second term is the light-induced spin coupling. The light beam intensity and detuning from resonance are set such that the light-induced coupling frequency $\omega = 2\pi \times 1.98(1)$ MHz largely exceeds the Larmor precession frequency $\omega_{\mathrm{L}} = 2\pi \times 31.7(5)$ kHz. In such a regime the Hamiltonian of Eq. (1) takes the form of the so-called one-axis twisting Hamiltonian, originally introduced for generating spin squeezing[21,22,43]. We drive the spin dynamics using light pulses of duration $t \sim 10$ ns to 1. Once all laser fields are switched off, we perform a projective measurement of the spin along the $z$-axis in a Stern–Gerlach experiment (see Fig. 1c). Measuring the atom number corresponding to each projection value $m$ allows to infer the projection probabilities $\Pi_m$, $-J \le m \le J$.

**Quantum state collapses and revivals.** We first investigated the evolution of the spin projection probabilities $\Pi_m$ as a function of the light pulse duration $t$. As shown in Fig. 2, we find the spin dynamics to involve mostly the even $|m\rangle_z$ states. This behavior is expected from the structure of the $\hat{J}_x^2$ coupling, which does not mix the even-$|m\rangle_z$ and odd-$|m\rangle_z$ sectors.

Starting in $|-J\rangle_z$, we observe for short times that all even-$|m\rangle_z$ states get gradually populated. The magnetization $m_z \equiv \langle\hat{J}_z\rangle$ and spin projection variance $\Delta J_z^2$ relax to almost constant values $m_z = -0.3(2)$ and $\Delta J_z^2 = 33(1)$ in the whole range $0.2\pi < \omega t < 0.36\pi$. This behavior agrees with the expected collapse of coherence induced by a non-linear coupling. To understand its mechanism in our system, we write the initial state in the $x$ basis, as

$$|-J\rangle_z = \sum_m (-1)^m c_m |m\rangle_x, \quad c_m = 2^{-J}\sqrt{\binom{2J}{J+m}}. \qquad (2)$$

In this basis, the non-linear coupling $\hat{J}_x^2$ induces $m$-dependent phase factors, leading to the state

$$|\psi(t)\rangle = \sum_m (-1)^m e^{-im^2\omega t} c_m |m\rangle_x. \qquad (3)$$

The variations between the accumulated phase factors lead to an apparent collapse of the state coherence[46]. The collapse timescale $t_c$ can be estimated by calculating the typical relaxation time of the magnetization, yielding $t_c = 1/(\sqrt{2J}\omega)$,

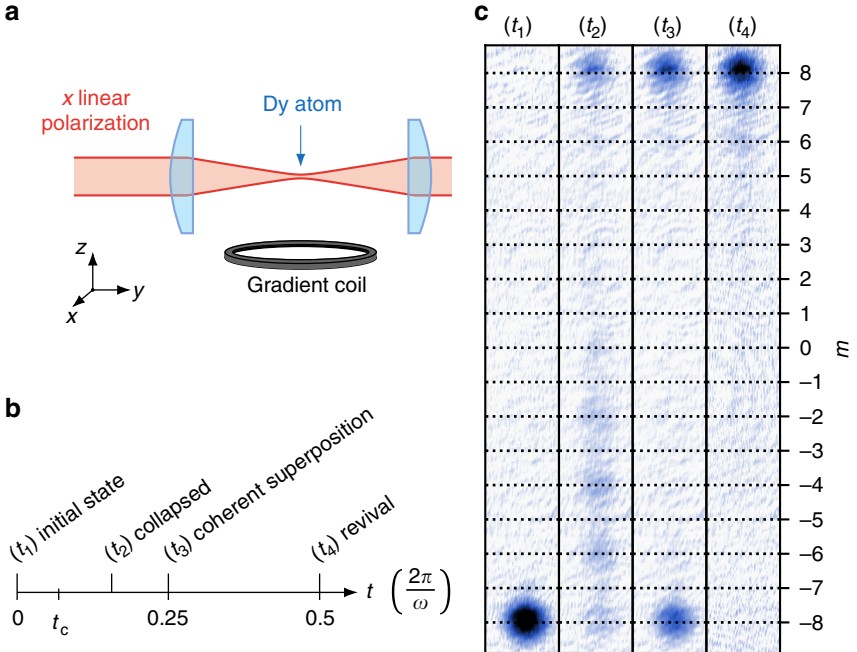

**Fig. 1** Experimental scheme and expected spin dynamics. **a** Experimental scheme. The spin $J = 8$ of Dy atoms is manipulated using an off-resonant laser field linearly polarized along $x$, leading to a non-linear coupling $\hbar\omega\hat{J}_x^2$. The spin state is subsequently probed by imaging the atoms after a Stern–Gerlach separation of magnetic sublevels $|m\rangle_z$, allowing to determine their individual populations. **b** Expected spin dynamics. The spin, initially prepared in $|-J\rangle_z$ (corresponding atom image in panel **c** for time $t_1$), first collapses to a featureless state (time $t_2$) on a fast timescale $t_c \ll 1/\omega$. We subsequently observe the formation of a superposition between states $|-J\rangle_z$ and $|J\rangle_z$ (time $t_3$) and later of the polarized state $|J\rangle_z$ (time $t_4$). Each image is the average of typically 10 resonant absorption images

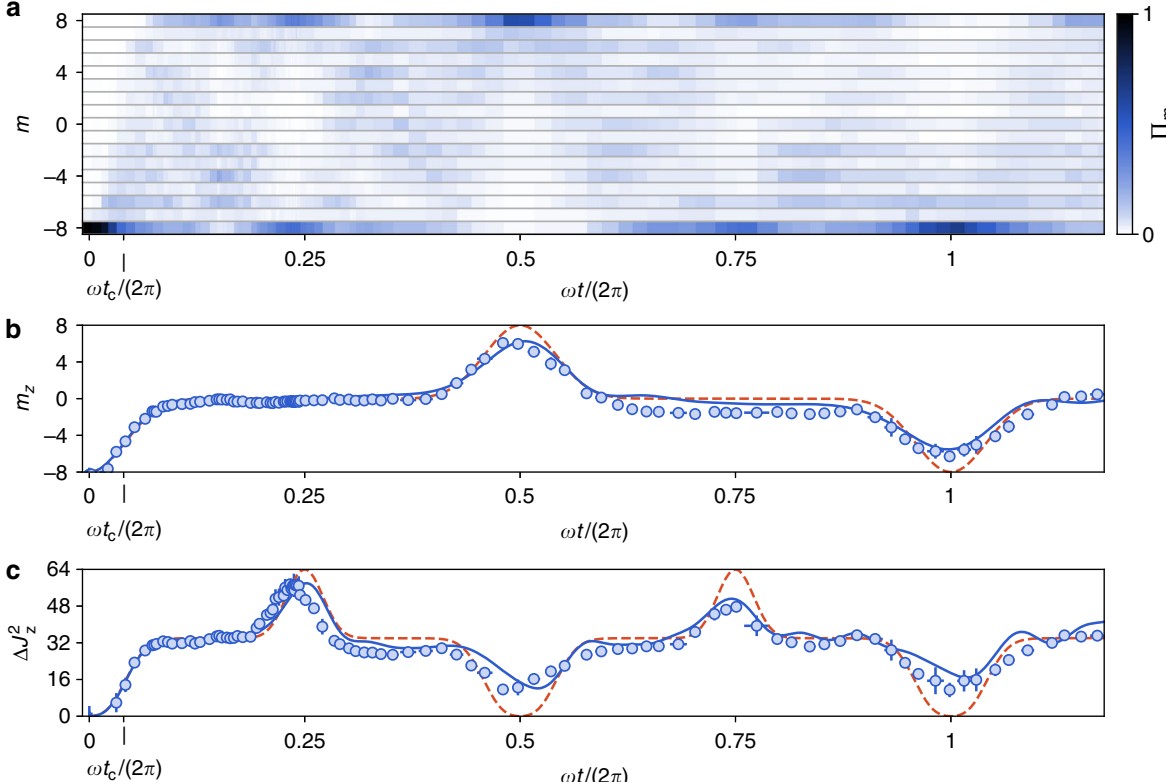

**Fig. 2** Collapses and revivals in the non-linear spin dynamics. **a** Evolution of the spin projection probabilities $\Pi_m$ along $z$ as a function of the light pulse area $\omega t$. **b**, **c** Evolution of the magnetization $m_z$ and of the variance in the spin projection $\Delta J_z^2$ calculated from the $\Pi_m$ distributions. The dashed red lines represent the ideal evolution for a $\hat{J}_x^2$ coupling[43], and the blue solid lines correspond to a fit taking into account experimental imperfections (see Methods). Each point is the average of five measurements, and the error bars represent the $1\sigma$ statistical uncertainty

i.e. $\omega t_c = 0.08\pi$[37,43] (see the Supplementary Note 1 and Supplementary Fig. 1).

For longer evolution times, we observe the occurence of peaks in $m_z(t)$ or $\Delta J_z^2$, that we interpret as the formation of states with significant quantum coherence[18,47,48]. After a quarter of the period, i.e. $\omega t = \pi/2$, all odd-$m$ (and all even-$m$) phase factors in Eq. (3) get in phase again, leading to the superposition

$$|\psi_{\text{kitten}}\rangle = e^{i\pi/4}(|-J\rangle_z - i|J\rangle_z)/\sqrt{2}, \qquad (4)$$

between maximally polarized states of opposite orientation[35,37], that we refer to as a 'kitten' state[14]. We observe that, for durations $0.45\pi < \omega t < 0.49\pi$, the magnetization remains close to zero while the variance in the spin projection features a peak of maximal value $\Delta J_z^2 = 57.1(2)$ (see Fig. 2).

For pure quantum states, such a large variance is characteristic of coherent superpositions between states of very different magnetization. However, from this sole measurement we cannot exclude the creation of an incoherent mixture of $|\pm J\rangle_z$ states. We observe at later times revivals of magnetization that provide a first evidence that the state discussed above indeed corresponds to a coherent quantum superposition. The first revival occurs around $\omega t = \pi$, and corresponds to a re-polarization of the spin up to $m_z = 6.0(1)$, with most of the atoms occupying the state $|J\rangle_z$. We detect another revival of magnetization around $\omega t = 2\pi$, corresponding to a magnetized state close to the initial state ($m_z = -6.0(2)$). Between these two revivals, we observe another superposition state (large spin projection variance $\Delta J_z^2 = 47.0(6)$) around $\omega t = 3\pi/2$.

The observed spin dynamics qualitatively agrees with the one expected for a pure $\hat{J}_x^2$ coupling[43] (dashed red line in Fig. 2), while a more precise modeling of the data—taking into account the linear Zeeman coupling produced by the applied magnetic field, as well as a fit of experimental imperfections (see Methods)—matches well our data (blue line in Fig. 2).

**Probing the coherence of the superposition**. In order to directly probe the coherences we follow another experimental protocol allowing us to retrieve the spin projection along directions lying in the $xy$ equatorial plane, corresponding to observables $\hat{J}_\phi \equiv \cos\phi \hat{J}_x + \sin\phi \hat{J}_y$ (see Methods). The coherence of the state $|\psi_{\text{kitten}}\rangle$, involving the opposite coherent states $|\pm J\rangle_z$, cannot be probed using a linear spin observable, such as the magnetization, but requires interpreting the detailed structure of the probability distributions $\Pi_m(\phi)$[49]. By expanding the coherent states $|\pm J\rangle_z$ on the eigen-basis $|m\rangle_\phi$ of the spin component $\hat{J}_\phi$, we rewrite the state as

$$|\psi_{\text{kitten}}\rangle = \frac{e^{i\pi/4}}{\sqrt{2}} \sum_m \left[ e^{-i(J\phi + m\pi)} - e^{i\left(J\phi + \frac{\pi}{2}\right)} \right] c_m |m\rangle_\phi \qquad (5)$$

where the $c_m$ coefficients were introduced in Eq. (2). For the particular angles $\phi = (p + 1/4)\pi/J$ ($p$ integer), the two terms in brackets cancel each other for odd $m$ values. Alternatively, for angles $\phi = (p - 1/4)\pi/J$ we expect destructive interferences for even $m$[8,49]. This behavior can be revealed in the parity of the spin projection

$$P(\phi) \equiv \sum_m (-1)^m \Pi_m(\phi) = \sin(2J\phi), \qquad (6)$$

which oscillates with a period $2\pi/(2J)$.

As shown in Fig. 3a, the experimental probability distributions $\Pi_m(\phi)$ feature strong variations with respect to the angle $\phi$. The center of mass of these distributions remains close to zero, consistent with the zero magnetization of the state $|\psi_{\text{kitten}}\rangle$. We furthermore observe high-contrast parity oscillations agreeing

with the above discussion and supporting quantum coherence between the $|\pm J\rangle_z$ components (see Fig. 3c).

Information on maximal-order coherences can be unveiled using another measurement protocol, which consists in applying an additional light pulse identical to the one used for the kitten state generation[50]. When performed right after the first pulse, the second pulse brings the state $|\psi_{\text{kitten}}\rangle$ to the polarized state $|J\rangle_z$, which corresponds to the second revival occuring around $\omega t = \pi$ in Fig. 2. An additional wait time between the two pulses allows for a Larmor precession of angle $\phi$ around $z$, leading to the expected evolution

$$|\psi(\phi)\rangle = \cos(J\phi)|J\rangle_z + \sin(J\phi)|-J\rangle_z, \qquad (7)$$

$$m_z(\phi) = J\cos(2J\phi). \qquad (8)$$

We vary the wait time and measure corresponding probability distributions $\Pi_m(\phi)$ (Fig. 3b) and magnetization $m_z(\phi)$ (Fig. 3c) consistent with Eqs. (7) and (8), respectively. This non-linear detection scheme reduces the sensitivity to external perturbations, as it transfers information from high-order quantum coherences onto the magnetization, much less prone to decoherence. It also decreases the requirements on the detection noise[51–55].

**A highly sensitive one-atom magnetic probe**. The Larmor precession of the atomic spins in small samples of atoms can be used for magnetometry combining high spatial resolution and high sensitivity[56]. While previous developments of atomic magnetometers were based on alkali atoms, multi-electron lanthanides, such as erbium or dysprosium intrinsically provide an increased sensitivity due to their larger magnetic moment, and potentially a substantial quantum enhancement when probing with non-classical spin states[57].

We interpret below the oscillation of the parity $P(\phi)$ discussed in the previous section as the footing of a magnetometer with quantum-enhanced precision, based on the non-classical character of the kitten state. According to generic parameter estimation theory, the Larmor phase $\phi$ can be estimated by measuring a generic observable $\hat{\mathcal{O}}$ with an uncertainty

$$\Delta\phi = \frac{\Delta\hat{\mathcal{O}}}{d\langle\hat{\mathcal{O}}\rangle/d\phi} \qquad (9)$$

for a single measurement[58]. Measuring the angle $\phi$ using coherent spin states (e.g. in a Ramsey experiment) leads to a minimum phase uncertainty $\Delta\phi_{\text{SQL}} = 1/\sqrt{2J}$, corresponding to the standard quantum limit (SQL). For an uncertainty limit on phase measurement $\Delta\phi$ we define the metrological gain compared to the SQL as the ratio $G \equiv (\Delta\phi_{\text{SQL}}/\Delta\phi)^2$, also commonly referred to as the quantum enhancement of measurement precision[5]. In this framework, the parity oscillation $P(\phi)$ expected from Eq. (6) for the state $|\psi_{\text{kitten}}\rangle$ yields a metrological gain $G = 2J$, corresponding to the best precision limit $\Delta\phi = 1/(2J)$ achievable for a spin $J$—the Heisenberg limit. From the finite contrast $C = 0.74(2)$ of a sine fit of the measured parity oscillation, we deduce a metrological gain $G = 2JC^2 = 8.8(4)$.

A further increase of sensitivity can be achieved using the full information given by the measured probability distributions $\Pi_m(\phi)$ (see Fig. 3a), i.e. without assuming the parity to be the most sensitive observable to measure phase variations[44]. In this more general approach, the phase sensitivity is obtained from the rate of change of the probability distribution $\Pi_m(\phi)$ upon a variation of $\phi$, that we quantify using the Hellinger distance $d_H^2(\phi, \phi') \equiv \frac{1}{2}\sum_m \left[\sqrt{\Pi_m(\phi)} - \sqrt{\Pi_m(\phi')}\right]^2$ between the

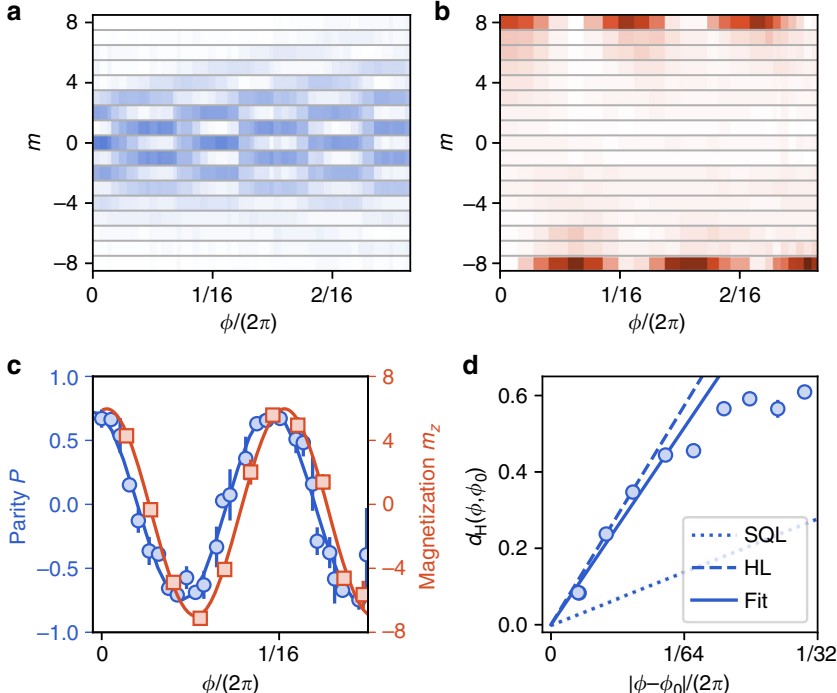

**Fig. 3** Probing quantum coherence. **a** Measured spin projection probabilities $\Pi_m(\phi)$ on the equatorial directions of azimutal angle $\phi$. **b** Measured spin projection probabilities $\Pi_m(\phi)$ along $z$, after a Larmor rotation of angle $\phi$ followed by a light pulse inducing further non-linear spin dynamics. **c** Evolution of the mean parity $P$ (blue circles) and magnetization $m_z$ (red squares) calculated from the probabilities shown in **a** and **b**, respectively. The phase shift between the measured oscillations stems from the phase offset associated with the Larmor rotation around $z$ occurring during the parity measurement sequence. **d** Variation of the Hellinger distance between projection probabilities of angles $\phi$ and $\phi_0$ as a function of the relative angle $\phi - \phi_0$, calculated from the data plotted in **a** with $\phi_0 = 0.02$. The solid line corresponds to the linear variation for small angle differences fitted to the data. The dotted (dashed) line corresponds to the standard quantum limit (Heisenberg limit). Error bars represent the $1\sigma$ statistical error

distributions $\Pi_m(\phi)$ and $\Pi_m(\phi')$. For small angle differences, one expects the scaling behavior $d_H(\phi, \phi') \simeq \sqrt{F/8}|\phi - \phi'|$, where $F$ is the classical Fisher information, which quantifies the measurement sensitivity as $\Delta\phi = 1/\sqrt{F}$[44,58]. For coherent spin states, the Fisher information $F = 2J$ corresponds to a measurement precision at the SQL. More generally, an increase in the slope of the Hellinger distance variation signals a gain in precision compared to the SQL, quantified by the metrological gain $G = F/(2J)$. For the kitten state given by Eq. (5), we expect a metrological gain $G = 2J$ at the Heisenberg limit. We show in Fig. 3d the Hellinger distance computed from the distributions $\Pi_m(\phi)$ shown in Fig. 3a. Its variation for small angle differences yields a metrological gain $G = 13.9(1.1)$. We thus find that using the full information from the probability distributions—rather than using its parity $P(\phi)$ only—increases the phase sensitivity.

For a given quantum state used to measure the Larmor phase, we expect the metrological gain to remain bounded by the value of its spin projection variance, as $G \leq 2\Delta J_z^2/J = 14.3(1)$[58]. As the measured gain coincides with this bound within error bars, we conclude that the phase measurement based on the Hellinger distance is optimum. We also performed a similar Hellinger distance analysis based on the distributions $\Pi_m(\phi)$ shown in Fig. 3b leading to a comparable metrological gain $G = 14.0(9)$ (see the Supplementary Note 3). Further increase of sensitivity would require improving the state preparation.

**Tomography of the superposition state**. In order to completely characterize the superposition state, we perform a tomographic reconstruction of its density matrix[59]. The latter involves $(2J + 1)^2 - 1 = 288$ independent real coefficients, that we

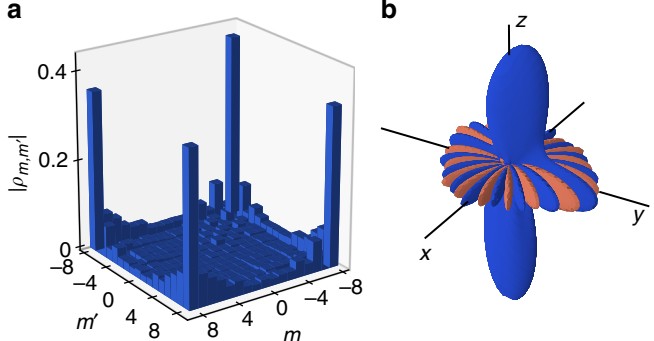

**Fig. 4** Tomographic reconstruction of the kitten state. **a** Absolute values of the density matrix coefficients $|\rho_{m,m'}|$ fitted from spin projection measurements performed along $z$ and in the equatorial plane. **b** Angular Wigner function corresponding to the density matrix plotted in **a**. Negative-valued regions are plotted in red

determine from a fit of the spin projection probabilities $\Pi_m$ measured on the $z$-axis and on a set of directions uniformly sampling the $xy$ equatorial plane[60]. The inferred density matrix is plotted in Fig. 4a. Its strongest elements correspond to populations and coherences involving the coherent states $|\pm J\rangle_z$, as expected for the state $|\psi_{\text{kitten}}\rangle$. We measure a coherence to population ratio $2|\rho_{-J,J}|/(\rho_{-J,-J} + \rho_{J,J}) = 0.92(8)$.

In order to further illustrate the non-classical character of the superposition state, we compute from the density matrix its associated Wigner function $W(\theta, \phi)$[22], defined for a spin over the

spherical angles $\theta$, $\phi$ as

$$W(\theta, \phi) = \sum_{\ell=0}^{2J} \sum_{m=-\ell}^{\ell} \rho_\ell^m Y_\ell^m(\theta, \phi), \qquad (10)$$

where $\rho_\ell^m$ is the density matrix component on the spherical harmonics $Y_\ell^m(\theta, \phi)$[61]. The reconstructed Wigner function, plotted in Fig. 4b, exhibits two lobes of positive value around the south and north poles, associated with the population of the states $|\pm J\rangle_z$. It also features interferences around the equatorial plane originating from coherences between these two states, with strongly negative values in a large phase space area. This behavior directly illustrates the highly non-classical character of the kitten state.

**Dephasing due to classical noise.** We furthermore investigated the environment-induced decay of quantum coherence by following the evolution of density matrices $\rho(t)$ reconstructed after variable wait times $t$ in the 10–100 μs range.

While we do not detect significant evolution of the populations $\Pi_m$, we observe a decrease of the extremal coherence $|\rho_{-J,J}|$, of 1/e decay time $\tau = 58 \pm 4$ μs, which we attribute to fluctuations of the ambient magnetic field. To calibrate such a dephasing process, we study the damping of the amplitude $J_\perp(t)$ of a coherent state, initially prepared in the state $|J\rangle_x$ and evolving under the applied magnetic field along $z$ and the ambient magnetic field fluctuations (see Methods). As shown in Fig. 5b, the transverse spin amplitude $J_\perp$ decays on a 1/e timescale $\tau_0 = 740 \pm 80$ μs, consistent with residual magnetic field fluctuations in the mG range. The decoherence rate of the kitten state is thus enhanced by a factor $\tau_0/\tau = 13(2)$ compared to a coherent state, which illustrates the intrinsic fragility of mesoscopic coherent superpositions.

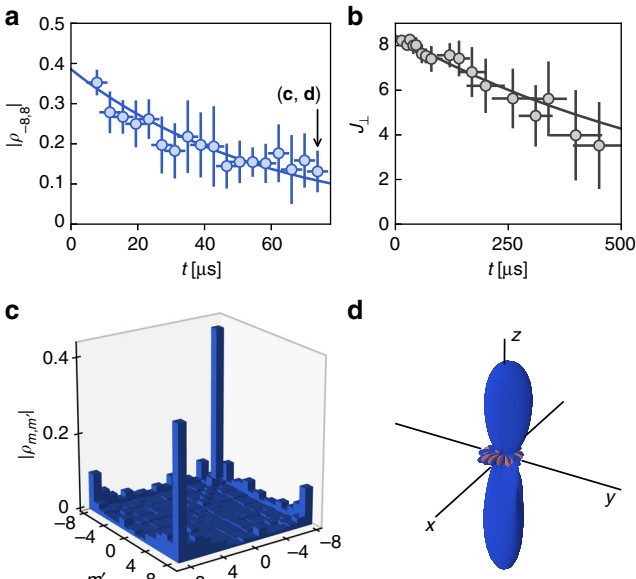

**Fig. 5** Dephasing of coherences. **a** Evolution of the modulus of the extremal coherence $|\rho_{-8,8}|$ (blue circles) calculated from the tomography of the superposition state after a wait time $t$. The horizontal error bars correspond to the standard deviation of the Larmor precession times required for tomography. Vertical error bars are the $1\sigma$ statistical error computed using a random-weight bootstrap method. **b** Evolution of the mean transverse spin amplitude $J_\perp$ for an initial state $|J\rangle_x$ in the same magnetic field environment than for the data in **a**. The solid lines in **a** and **b** are exponential fits of the data. **c**, **d** Density matrix and Wigner function reconstructed for $t = 70 \pm 3$ μs, i.e. after a strong damping of coherences

Spin decoherence due to magnetic field fluctuations can be modeled similarly to the $T_2^*$ decay in nuclear magnetic resonance[62] (see the Supplementary Note 4). Using a magnetic probe located close to the atom position, we measure shot-to-shot magnetic field fluctuations on a 0.5-mG range, but their variation on the ~100-μs dephasing timescale remains negligible. In this regime, we expect the dephasing of the state $|\psi_{kitten}\rangle$ to occur $2J = 16$ times faster than for a coherent state, a value close to our measurement.

Finally, we plot in Fig. 5c, d the reconstructed density matrix and its associated Wigner function for the wait time $t = 70 \pm 3$ μs. The weak amplitude of coherences and the shrinking of the negative regions in the Wigner function illustrate the dynamics towards an incoherent statistical mixture[6].

**Discussion**
In this work, we use spin-dependent light shifts to drive the electronic spin $J = 8$ of dysprosium atoms under a non-linear one-axis twisting Hamiltonian. The observation of several collapses and revivals of quantum coherence shows that the spin dynamics remains coherent over a full period of the evolution. In particular, the state produced after one quarter of the period consists of a coherent superposition between quasi-classical spin states of opposite orientation, which can be viewed as a mesoscopic instance of Schrödinger cat. While such coherent dynamics could be achieved with individual alkali atoms of smaller spin size[39,40], the realization of large-size coherent superpositions with ensembles of spin-1/2 particles is extremely challenging[9,17]. The high fidelity of our protocol stems from the reduced size $2J + 1$ of the available Hilbert space, that scales linearly with the effective distance $2J$ between the states involved in the superposition. Such scaling contrasts with the exponential scaling in the number of accessible states for ensembles of qubits, which dramatically increases the number of decoherence channels. Similarly, the full tomographic reconstruction of the produced quantum state also crucially relies on this limited size of the Hilbert space. Quantum state tomography of an equivalent 16-qubit ensemble remains inaccessible, unless restricting the Hilbert space to the permutationally invariant subspace[63] or using compressed sensing for almost pure states[64].

We show that our kitten state provides a quantum enhancement of precision of 13.9(1.1), up to 87(2)% of the Heisenberg limit. So far, such a high value could only be reached in ensembles of thousands of qubits based on multiparticle entanglement[25,27–30]. In such systems, while entanglement occurs between a large number of qubits, the quantum enhancement of precision remains small compared to the system size, far from the Heisenberg limit. Our protocol could be extended to prepare kitten states $(|-K\rangle_z - i|K\rangle_z)/\sqrt{2}$ with $|K| \le J$, by initiating the atoms in $|-K\rangle_z$ before applying a non-linear spin coupling identical to the one used in this work. This would allow us to demonstrate the Heisenberg scaling of measurement sensitivity $\delta\phi \propto 1/K$. We could also implement, using similar techniques, protocols to prepare non-classical states based on adiabatic evolutions[65–67].

Our method could also be applied to systems of larger electronic spin $J$. Dysprosium being the optimum choice among all atomic elements in the electronic ground state, further improvement would require using highly excited electronic levels, such as Rydberg atomic states[12], or using ultracold molecules[68]. By increasing the atom density, one could also use interactions between $N$ atoms of spin $J$ to act on a collective spin of very large size $\mathcal{J} = NJ$, allowing to explore non-classical states of much larger size.

**Methods**
**Sample preparation and detection.** We use samples of about $9(1) \times 10^4$ atoms of $^{164}$Dy, cooled to a temperature $T \simeq 2$ μK using laser cooling and subsequent

evaporative cooling in an optical dipole trap[69]. The dipole trap has a wavelength $\lambda = 1064$ nm, resulting in negligible interaction with the atomic spin[70]. The samples are initially spin-polarized in the absolute ground state $|-J\rangle_z$, with a bias field $B_z \simeq 0.5$ G along $z$, such that the induced Zeeman splitting largely exceeds the thermal energy. Before starting the light-induced spin dynamics, we ramp the bias field down to the final value $B_z = 18.5(3)$ mG in 20 ms. We checked that the promotion to higher spin states (with $m > -J$) due to dipole–dipole interactions remains negligible on this timescale. The optical trapping light is switched off right before the spin dynamics experiments.

After the light-induced spin dynamics, we perform a Stern–Gerlach separation of the various spin components using a transient magnetic field gradient (typically 50 G/cm during 2 ms) with a large bias magnetic field along $z$. After a 3.5 ms time of flight, the atomic density is structured as 17 separated profiles (see Fig. 1c), allowing to measure the individual spin projection probabilites $\Pi_m$ using resonant absorption imaging, where $m$ is the spin projection along $z$. The relative scattering cross-sections between $|m\rangle_z$ sub-levels are calibrated using samples of controlled spin composition.

Spin projection measurements along equatorial directions are based on spin rotations followed by a projective measurement along $z$. We apply a magnetic field pulse along $y$, of temporal shape $B_y(t) = B_y^{max} \sin^2(\pi t/\tau)$, with $\tau = 3$ and $B_y^{max}$ adjusted to map the $z$-axis on the equator. Taking into account the static field along $z$, we expect the pulse to map the equatorial direction of azimutal angle $\phi_i \simeq 0.35$ rad on the $z$-axis. An arbitrary angle $\phi = \phi_i + \phi_L$ can be reached using an additional wait time before the $B_y$ pulse, allowing for a Larmor precession of angle $\phi_L$. The calculation of the angle $\phi_L$ uses the magnetic field component $B_z$ measured using an external probe, allowing to reduce the effect of shot-to-shot magnetic field fluctuations.

**Spin dynamics modeling**. Quantitative understanding of the observed spin dynamics requires taking into account experimental imperfections. We include the linear Zeeman coupling induced by the magnetic field applied along $z$ (see Eq.(1)), leading to a small Larmor rotation on the typical timescales used for the light-induced spin dynamics. We also take into account the slight polarization ellipticity expected from the focusing of the laser beam on the atomic sample (beam divergence $\theta = \lambda/(\pi w) \simeq 4$ mrad). Finally, we improve the spin dynamics modeling by fitting a small angle mismatch $\simeq 8°$ between the quantization field and the $z$-axis. More details on this modeling can be found in the Supplementary Note 2.

**Quantum state tomography**. The density matrix of the kitten state is determined from a least-square fit of the measured spin projection probabilities $\Pi_m$ along $z$ and $\Pi_m(\phi)$ on equatorial directions[60]. We uniformly sample the equatorial plane using a set of azimutal angles $\phi \in [\phi_0, \phi_0 + \pi]$. The procedure thus requires variable spin rotation durations (on average $\simeq 10$ μs), which limits the quality of the tomography due to dephasing. To reduce its effect, we use the magnetic field values measured for each experiment with an external probe to compensate for part of the dephasing, which increases the quality of the tomography and extents the coherence times by a factor $\simeq 3$. The robustness of the method with respect to measurement noise and finite sampling is tested using a random-weight bootstrap method, from which we define the statistical error bars in Fig. 5.

**Calibration of dephasing**. To calibrate the dephasing of coherences due to magnetic field fluctuations, we perform a Ramsey experiment using coherent spin states. We start in the ground state $|-J\rangle_z$ that we bring on the equator using a $\pi/2$ magnetic field pulse applied along $y$. We then let the spin precess around $z$ for a duration $t$, and subsequently perform a second $\pi/2$ pulse before performing a spin projection measurement along $z$. We observe Ramsey oscillations of the magnetization $m_z(t) = J_\perp(t)\cos(\omega_L t + \phi)$, where the local oscillation contrast $J_\perp(t)$ corresponds to the transverse spin amplitude shown in Fig. 5b.

## Data availability

The datasets generated and analyzed during the current study are available from the corresponding author on request.

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

## Acknowledgements

This work is supported by PSL University (MAFAG project) and European Union (ERC UQUAM and TOPODY, Marie Curie project 661433). We thank F. Gerbier, R. Lopes, and P. Zoller for fruitful discussions.

## Author contributions

T.C., L.S., C.B., A.E., V.M., and D.D. carried out the experiment. J.D. and S.N. supervised the project. All authors contributed to the discussion, analysis of the results, and the writing of the manuscript.

## Additional information

**Competing interests:** The authors declare no competing interests.

