## [Peer Review File · Nature Communications]

Reviewers' comments:

Reviewer #1 (Remarks to the Author):

In the manuscript "Quantum-enhanced sensing using non-classical spin states of a highly magnetic atom", Chalopin et al. report on the experimental realization of a non-linear light-spin interaction able to drive spin $J=8$ single atoms initialized in $|-J\rangle$ to the macroscopic superposition state $|+J\rangle + |-J\rangle$. They first observe collapses and revivals of the non-linear evolution: a strong indication of coherent dynamics. Then, they focus the analysis on the state $|+J\rangle + |-J\rangle$, showing high quantum coherence and enhanced sensitivity (over the bound sets by coherent spin states) to magnetic fields. They further perform a tomographic reconstruction of the state and analyze the decay of off-diagonal terms in presence of dephasing.

I really enjoyed reading this manuscript. The results are important, timely and, in my opinion, of high impact. The presentation is neat. Data and figures are clear. I hope that my comments below may help the authors to further improve and clarify some aspects of their manuscript.

Introduction:

- My personal impression is that the introduction is a little hazy regarding the notion macroscopic superposition states of a single spin- J atom (a system living in a $2J+1$ dimensional space) and that of a collection of N spin- $1/2$ atoms (realizing an effective spin- $N/2$ and living in a 2^N dimensional space). To avoid any confusion, the authors should clarify the difference between the two systems: for instance, the sentence "Each atomic spin ..." comes a little too late in the discussion.
- The authors cite reference [33] for coherent spin states. This is correct, but I would also add Arecchi et al PRA 6, 2211 (1972), that is a more standard reference for coherent spin states
- In the sentence "while spin-squeezed states have been realized experimentally [21–27, 30]" the correct list of reference here is [21-23,26,27,29-31] (Ref. [23] has realized reduced atom-number fluctuations rather than spin squeezing). They should also consider citing Bohnet et al., Science 352, 1297 (2016) regarding the creation of spin-squeezed states of trapped ions.
- In the sentence "cat states remain out of reach ..." the authors should add the reference [53] as this work is the first, to the best of my knowledge, that has demonstrated entanglement beyond spin squeezing in these systems.

Results

- The authors should add one or two sentences about what generates the non-linear spin term in the spin Hamiltonian (1). At present, the authors tell "The non-linear spin dynamics is induced by a laser beam focused on the atomic sample." This is not sufficient.
- The authors introduce the notion of Schrodinger kitten states: "Considering the spin $J = 8$ as a mesoscopic quantum system, we refer to this state as a Schrodinger 'kitten' state [15]." When a kitten is considered to be a cat? I fully understand that the reference [15] talks about a kitten. However, from Fig.1 c3 I clearly see a cat. I suggest to remove "kitten" everywhere in the text but, if the authors want to be conservative, the notion of "kitten" should definitely appear and be clarified in the introduction (see my comment above about hazy notion of macroscopic superposition states).
- In Fig.1 I would call c4 the first revival, rather than c3; and c3 the time of formation of the cat: I do not understand the formation of the cat as a revival (see Fig.2).
- It is not clear if the blue line in Fig.2 b and c is a fit or a theoretical model. The Method section is

of no help. I would like to understand better what is this line. At the moment, I would not be able to reproduce it.

- The operator O in Eq.(9) is not definite.
- Below Eq.(9) there is a very interesting result that is not emphasized enough, I think. The standard analysis of cat states is done in term of parity oscillations. The authors measure a gain factor $G=8.8$ that corresponds to the normalized Fisher information (see for instance PNAS 113, 11459) $G=F_{\text{parity}}/(2*J)=C^2*(2*J)$, where C is the visibility (or contrast) of the parity fringe. The authors can obtain this result also via the Hellinger method $d_H^2 = \frac{1}{2}*\sum_p (\sqrt{P(p|\phi)} - \sqrt{P(p|\pi/2)})^2 \sim F_{\text{parity}}/8 (\phi-\pi/2)^2$, where $P(p|\phi)$ is the probability to observe the parity $p=+/-1$ at the phase angle ϕ (the reference angle for the calculation of the Hellinger distance is $\pi/2$ here). I would expect this to be the best strategy for cat states.

Yet, the authors measure the magnetization and calculate the Hellinger distance from the corresponding probability distribution. They obtain the Fisher information F_{magnet} from a the fit of the Hellinger distance and find $F_{\text{magnet}}=13.9*(2*J)$, larger than $F_{\text{parity}}=8.8*(2*J)$, even taking into account errorbars. This result is very surprising and interesting. To me it looks correct (the method is correct and used elsewhere and I can easily confirm the result by looking at figure 3d).

The result $F_{\text{magnet}} > F_{\text{parity}}$ seems to suggest that parity is more fragile than the magnetization in presence of experimental noise. This result should be emphasized more: First, I would put in Fig 3d data for the Hellinger distance calculated as discussed above for the parity signal (to see the different slope). Then, I would like to confirm this result within a theoretical calculation: the authors seem to have a theoretical model of their experiment including noise (see solid line in Fig.2), it might not be difficult to obtain from this model the theoretical parity and magnetization probabilities, and then the corresponding Fisher informations. This would provide an understanding why (surprisingly, to me) the magnetization gives a higher Fisher information.

Again, this is a notable results since many experiments that have created cat states so far (in ion experiments, for instance) have analyzed the signal in terms of parity oscillations: it looks like this is not the best strategy, at least here.

In conclusion, I have no doubts that this manuscript deserves the publication in Nature Communications. Nevertheless, before recommending the publication, I ask the authors to take into account my comments.

Reviewer #2 (Remarks to the Author):

Quantum superposition states of a mesoscopic object, such as the cat state, play a major role in quantum-enhanced metrology. However, due to the difficulty in control the mesoscopic quantum object and decouple it from the environment, it is still challenging to realize and manipulate the mesoscopic quantum superposition states. The authors used the highly magnetic Dysprosium atoms, with a mesoscopic spin size $J=8$, to generate the mesoscopic superposition state, verify its quantum coherences and use it as a probe for quantum-enhanced magnetic field sensing. The manuscript is well written and the results look sound. However, I have some concerns need to be addressed.

i) The quantum enhancement in the experiment comes from the large number of electric spin J , and J acts the role of particle number ($N=2J$) in the conventional quantum-enhanced

measurement. In the experiment, $J=8$ is fixed, and the corresponding precision of the kitten state ("cat state") is given, which is enhanced by a factor compared with SQL. However, in most of scenarios, the Heisenberg limit is a scaling of the measurement precision versus the particle number. Is it possible to prepare other kitten states $(|-K\rangle - i|K\rangle)/\sqrt{2}$ with different K (such as 7,6,5,...), and demonstrate the Heisenberg-limited scaling?

ii) In the experiment, how to resolve full information of the measured probability distributions? Fisher information is also an important quantity in metrology. The measure of classical Fisher information (CFI) also requires the probability distribution, is the result of Hellinger distance equivalent to the CFI?

iii) Two methods for extracting the phase are demonstrated, one is parity and the other is non-linear detection. Is this non-linear detection scheme similar to the proposals in Ref. [PRL 119, 193601 (2017), PRA 98, 012129 (2018), arXiv:1707.08260]? And what kind of external perturbations is almost immuned, detection noise? Please give more details on how to perform the non-linear detection and its advantages.

Reviewer #3 (Remarks to the Author):

This article reports brilliant experimental work in which the authors prepare and detect superposition states among the extremal Zeeman sublevels, $M = \pm 8$ of dysprosium atoms.

A cat-like superposition of the extremal Zeeman states is formed under the action of Larmor precession and a light shift term in the Hamiltonian. The state evolves in time and its components are measured by Stern-Gerlach separation, which together with spin rotations yield full tomographic information about the quantum states. It is shown that the states are more sensitive to magnetic fields than spin coherent states and that they decohere faster. These results are both shown to be in agreement with theoretical estimates.

The experimental details are described pedagogically and in convincing technical detail, and the theory elements are all very clear, and provide a perfect match to the experimental data. The article is testimony to the level of perfection mastered in the control and detection of atomic quantum states and dynamics.

The article is very well written and the relevant literature in this very active field of research is properly cited. For many readers it may be new information that by manipulating the internal ground state of real atoms, one may increase their sensitivity to magnetic fields in a more straightforward manner than by inducing entanglement and spin squeezing among the atoms. As such it constitutes a novel and valuable contribution to the already rich literature on quantum metrology.

I warmly recommend publication of the manuscript with no changes.

Dear referees,

We hereby resubmit our manuscript NCOMMS-18-18113, entitled ‘Quantum-enhanced sensing using non-classical spin states of a highly magnetic atom’.

We thank the referees for their careful reading of the paper and for their positive reviews. We have addressed in details all their recommendations, please see our answers below.

Sincerely,

The authors.

Reviewer #1

- *My personal impression is that the introduction is a little hazy regarding the notion macroscopic superposition states of a single spin- J atom (a system living in a $2J + 1$ dimensional space) and that of a collection of N spin- $1/2$ atoms (realizing an effective spin- $N/2$ and living in a 2^N dimensional space). To avoid any confusion, the authors should clarify the difference between the two systems: for instance, the sentence ‘Each atomic spin...’ comes a little too late in the discussion.*

We agree that the distinction between the single spin systems and the atomic ensembles could be emphasized more. For this, we added a sentence in the paragraph introducing cat states in single spin systems to mention the size of the Hilbert space: ‘We mention that the Hilbert space dimension of $2J + 1$ scales linearly with the separation between the two coherent states of the superposition.’ In the following paragraph introducing atomic ensembles, we added a sentence to stress the exponential scaling of the Hilbert space size with N , and its consequence on the robustness of cat states ‘This behavior results from the large size 2^N of the Hilbert space, which scales exponentially with the system size N , resulting in a large number of decoherence channels (e.g. losing a single particle fully destroys their quantum coherence) [47].’ In the next paragraph introducing our system, we stress again the difference between the single-spin $J = 8$ and the equivalent ensemble of 16 spins $1/2$ in terms of size of the Hilbert space. This should remove any confusion between the two types of systems.

- *The authors cite reference [33] for coherent spin states. This is correct, but I would also add Arecchi et al PRA 6, 2211 (1972), that is a more standard reference for coherent spin states.*

We agree that this reference is relevant and we added it after the reference [33] (appearing as [35] in the new version of the manuscript).

- *In the sentence ‘while spin-squeezed states have been realized experimentally [21-27, 30]’ the correct list of reference here is [21-23,26,27,29-31] (Ref. [23] has realized reduced atom-number fluctuations rather than spin squeezing). They should also consider citing Bohnet et al., Science 352, 1297 (2016) regarding the creation of spin-squeezed states of trapped ions.*

We agree with the proposition of the referee and modified the sentence as suggested, including the additional reference to Bohnet et al., Science 352, 1297 (2016).

- *In the sentence ‘cat states remain out of reach...’ the authors should add the reference [53] as this work is the first, to the best of my knowledge, that has demonstrated entanglement beyond spin squeezing in these systems.*

We modified this sentence to refer to [53] (now appearing as [46]) after the list of references to spin squeezing: ‘ In such systems, spin-squeezed states have been realized experimentally [21-23,26,27,29-32], as well as non-gaussian entangled states [46]. ’

- *The authors should add one or two sentences about what generates the non-linear spin term in the spin Hamiltonian (1). At present, the authors tell ‘The non-linear spin dynamics is induced by a laser beam focused on the atomic sample.’ This is not sufficient.*

We now give more details on the origin of the spin-dependent light shifts, as follows.

‘The non-linear spin dynamics results from spin-dependent energy shifts induced by a laser beam focused on the atomic sample. The laser wavelength is chosen close to the 626-nm resonance line, such that the light shifts are proportional to the polarizability tensor of a $J = 8$ to $J' = 9$ optical transition. For a linear light polarization along x , the light shift operator reduces to a coupling $\propto J_x^2$ (up to a constant), and we expect the spin dynamics to be described by the Hamiltonian [40]

$$\hat{H} = \hbar\omega_L \hat{J}_z + \hbar\omega \hat{J}_x^2, \quad (1)$$

where the first term corresponds to the Larmor precession induced by the magnetic field, and the second term is the light-induced spin coupling.’

- *The authors introduce the notion of Schrodinger kitten states: ‘Considering the spin $J = 8$ as a mesoscopic quantum system, we refer to this state as a Schrodinger ‘kitten’ state [15].’ When a kitten is considered to be a cat? I fully understand that the reference [15] talks about a kitten. However, from Fig.1 c3 I clearly see a cat. I suggest to remove ‘kitten’ everywhere in the text but, if the authors want to be conservative, the notion of “kitten” should definitely appear and be clarified in the introduction (see my comment above about hazy notion of macroscopic superposition states).*

We agree that the notion of kitten state comes late in the manuscript, which maintains an ambiguity on the macroscopic nature of the quantum superposition. We believe that our system size can be considered as large, but not macroscopic, and we prefer the conservative option of the name ‘kitten’, first introduced in the reference [48]. We added a new sentence at the end of the introduction to motivate explicitly this choice and introduce the reference [48], as follows. ‘As this size can be considered large, but not macroscopic according to the original Schrödinger idea, we will hereafter refer to such quantum superpositions as Schrödinger kitten states [48].’

- *In Fig.1 I would call c4 the first revival, rather than c3; and c3 the time of formation of the cat: I do not understand the formation of the cat as a revival (see Fig.2).*

We agree that, to be precise, the term revival should be used for the appearance of coherent states only. We no longer refer to c3 as a revival, as suggested by the referee. We

modified the Fig.1 and its legend accordingly, as well as the paragraph describing the collapse and revival dynamics.

- *It is not clear if the blue line in Fig.2 b and c is a fit or a theoretical model. The Method section is of no help. I would like to understand better what is this line. At the moment, I would not be able to reproduce it.*

The blue line is a fit to the data based on a model of the spin dynamics that includes experimental imperfections. The residual magnetic field and the small polarization ellipticity expected from the laser beam focusing are known and are taken into account in this model. We also fit an unknown angle mismatch between the quantization field and the z axis.

We have expanded the Methods section relative to this model, including the value of the laser beam divergence leading to the polarization ellipticity and the fitted angle between the quantization field and the z axis. We still refer to the Supplementary Materials for a complete description of the model, that we find too technical to appear in the main article.

We also modified the paragraph in the main text to refer more explicitly to the theory curves plotted in Fig.2 and to mention that the blue line is a more complex model including a fit of experimental imperfections. ‘The observed spin dynamics qualitatively agrees with the one expected for a pure \hat{J}_x^2 coupling [45] (dashed red line in Fig. 2), while a more precise modeling of the data – taking into account the linear Zeeman coupling produced by the applied magnetic field, as well as a fit of experimental imperfections (see Methods) – matches well our data (blue line in Fig. 2). ‘

- *The operator O in Eq.(9) is not definite.*

The relation given in Eq.(9) is valid for any operator \mathcal{O} . We thus added the adjective ‘generic’ to emphasize the generality of the argument.

- *Below Eq.(9) there is a very interesting result that is not emphasized enough, I think. The standard analysis of cat states is done in term of parity oscillations. The authors measure a gain factor $G = 8.8$ that corresponds to the normalized Fisher information (see for instance PNAS 113, 11459) $G = F_{\text{parity}}/(2J) = C^2(2J)$, where C is the visibility (or contrast) of the parity fringe. The authors can obtain this result also via the Hellinger method $d_H^2 = \frac{1}{2} \sum_p (\sqrt{P(p|\phi)} - \sqrt{P(p|\pi/2)})^2 \sim F_{\text{parity}}/8(\phi - \pi/2)^2$, where $P(p|\phi)$ is the probability to observe the parity $p = \pm 1$ at the phase angle ϕ (the reference angle for the calculation of the Hellinger distance is $\pi/2$ here). I would expect this to be the best strategy for cat states.*

The result $F_{\text{magnet}} > F_{\text{parity}}$ seems to suggest that parity is more fragile than the magnetization in presence of experimental noise. This result should be emphasized more: First, I would put in Fig 3d data for the Hellinger distance calculated as discussed above for the parity signal (to see the different slope). Then, I would like to confirm this result within a theoretical calculation: the authors seem to have a theoretical model of their experiment including noise (see solid line in Fig.2), it might not be difficult to obtain from this model the theoretical parity and magnetization probabilities, and then the corresponding Fisher informations. This would provide an understanding why (surprisingly, to me) the magnetization gives a higher Fisher information.

Again, this is a notable results since many experiments that have created cat states so far

(in ion experiments, for instance) have analyzed the signal in terms of parity oscillations: it looks like this is not the best strategy, at least here.

We believe that there was a misunderstanding on the data used to calculate the Hellinger distance, probably because it was specified in the caption of Fig.3 only. The Hellinger distances shown in Fig.3d are calculated using the full probability distributions of Fig.3a, i.e. the ones exhibiting the parity oscillations.

The parity oscillations shown in Fig. 3c could indeed be interpreted using the Hellinger distance formulas proposed by the referee. This analysis is almost equivalent to our analysis based on the contrast of the parity oscillations.

The Hellinger distances shown in Fig.3d are calculated differently, as we make use of the full probability distributions of Fig.3a and do not assume the parity to be the most suited observable. This more general approach allows to improve the sensitivity of the phase measurement, which shows that it is preferable to use the entire information available rather than using the parity oscillation only. The two procedures would lead to the same result in the absence of experimental imperfections, but we find here that using the Hellinger distance variations calculated over the full probability distributions significantly improves the phase sensitivity.

We emphasized which data is used for the calculation of the Hellinger distance, both in the legend of Fig.3 and in the main text. This should remove any ambiguity. We also modified the first sentence of the paragraph on the Hellinger distance analysis to stress the difference with the previous analysis of parity oscillations. ‘A further increase of sensitivity can be achieved using the full information given by the measured probability distributions $\Pi_m(\phi)$ (see Fig. 3a), i.e. without assuming the parity to be the most sensitive observable to measure phase variations [46].’

We can also calculate the Hellinger distance variations from the data of Fig.3b, that exhibits magnetization oscillations. We obtain from it a metrological gain of 14.0(9), to be compared to the value 13.9(1.1) obtained from the data showing parity oscillations. The two sets of data thus give almost identical results. This argument already appeared in the previous version of the manuscript.

Reviewer #2

- *The quantum enhancement in the experiment comes from the large number of electric spin J , and J acts the role of particle number ($N = 2J$) in the conventional quantum-enhanced measurement. In the experiment, $J = 8$ is fixed, and the corresponding precision of the kitten state (‘cat state’) is given, which is enhanced by a factor compared with SQL. However, in most of scenarios, the Heisenberg limit is a scaling of the measurement precision versus the particle number. Is it possible to prepare other kitten states $(|-K\rangle - i|K\rangle)/\sqrt{2}$ with different K (such as 7,6,5,...), and demonstrate the Heisenberg-limited scaling?*

The investigation of the Heisenberg scaling of sensitivity with K would be possible with a protocol similar to the one we studied for $K = J$. To prepare the kitten state $(|-K\rangle - i|K\rangle)/\sqrt{2}$, one should first prepare a sample polarized in the state $|J, m = -K\rangle$, and subsequently apply a non-linear coupling $\hbar\omega J_x^2$ for a quarter of the period, as for $K = J$. The preparation of spin-polarized samples in $|J, m = -K\rangle$ can be performed using adiabatic magnetic field sweeps in the presence of a light-induced quadratic Zeeman effect, as recently implemented

in the group of F. Ferlino (private communication). To measure the precision gain $G(K)$ with respect to the SQL for a given state $(|-K\rangle - i|K\rangle)/\sqrt{2}$, we could repeat the protocol corresponding to the data in Fig.3.

We added two sentences in the conclusion to describe this possible extension of our work, as follows. ‘Our protocol could be extended to prepare kitten states $(|-K\rangle - i|K\rangle)/\sqrt{2}$ with $|K| \leq J$, by initiating the atoms in $|-K\rangle$ before applying a non-linear spin coupling identical to the one used in this work. This would allow us to demonstrate the Heisenberg scaling of measurement sensitivity $\delta\phi \propto 1/K$.’

- *In the experiment, how to resolve full information of the measured probability distributions? Fisher information is also an important quantity in metrology. The measure of classical Fisher information (CFI) also requires the probability distribution, is the result of Hellinger distance equivalent to the CFI?*

The method we implemented to measure the phase sensitivity is inspired from the reference [46]. It quantifies the rate of variation of probability distribution using the Hellinger distance, giving access to the classical Fisher information. We agree that, given the importance of the concept of Fisher information in metrology, it should be mentioned explicitly. We modified the paragraph describing the phase sensitivity measurement, as follows. ‘For small angle differences, one expects the scaling behavior $d_H(\phi, \phi') \simeq \sqrt{F/8}|\phi - \phi'|$, where F is the classical Fisher information, which quantifies the measurement sensitivity as $\Delta\phi = 1/\sqrt{F}$ [46,60]. For coherent spin states, the Fisher information $F = 2J$ corresponds to a measurement precision at the SQL. More generally, an increase in the slope of the Hellinger distance variation signals a gain in precision compared to the SQL, quantified by the metrological gain $G = F/(2J)$. For the kitten state given by Eq. 5, we expect a metrological gain $G = 2J$ at the Heisenberg limit.’

- *Two methods for extracting the phase are demonstrated, one is parity and the other is non-linear detection. Is this non-linear detection scheme similar to the proposals in Ref. [PRL 119, 193601 (2017), PRA 98, 012129 (2018) , arXiv:1707. 08260]? And what kind of external perturbations is almost immuned, detection noise? Please give more details on how to perform the non-linear detection and its advantages.*

We agree that the advantages of non-linear detection scheme was only briefly explained in the previous version of the manuscript. We added a few references [53-57]. The references [55,56,57] proposed by the referee discuss the interest of a non-linear detection scheme to reduce the effect of detection noise and are indeed directly relevant here. We already mentioned that this scheme makes the detection less sensitive to decoherence. We added a new sentence on another advantage of this scheme with respect to detection noise, as follows. ‘This non-linear detection scheme reduces the sensitivity to external perturbations, as it transfers information from high-order quantum coherences onto the magnetization, much less prone to decoherence. It also decreases the requirements on the detection noise [53-57].’

Reviewer #3

We thank the referee 3 for his positive report.

REVIEWERS' COMMENTS:

Reviewer #1 (Remarks to the Author):

I thank the authors to have taken into account all my comments. I recommend this manuscript for publication in Nature Communications.

Reviewer #2 (Remarks to the Author):

According to the three reviewer reports, the authors have revised their manuscript. Most of their revisions are satisfactory. However, before accept for publication, the authors should clarify the following confusions and mention more related references.

(1) The authors should clarify the confusion on the relation between and a spin- $(N/2)$ system and an ensemble of N spin- $1/2$ atoms.

In lines 55-59, the authors mention a set of N spin- $1/2$ atoms behaves as an effective large spin of $J=N/2$, and the non-classical states can be prepared via interaction between atoms and cite Ref. [45].

In lines 63-68, the authors state that "this behavior results from the large size 2^N of the Hilbert space..."

Actually, for an ensemble of N identical bosons, which occupy two different internal spin states, they can be exactly mapped onto an effective large spin of $J=N/2$ and the corresponding Hilbert space is of $2J+1=N+1$ dimensions but not 2^N dimensions. For an example, see PRL 97, 150402 (2006).

The authors cite Ref. [45] for the preparation of non-classical states via inter-atom interactions. Actually, Ref. [45] only mentions the preparation of spin squeezing via time-evolution dynamics with fixed parameters. The authors should cite other related references on this topic, such as, the preparation of non-classical states via adiabatic evolutions (PRL 97, 150402 (2006); PRL 111, 180401 (2013); PRA 97, 032116 (2018)).

(2). In lines 54-55, the authors mention that non-classical spin states has been created in ensemble of one-electron atoms.

Actually, in addition to one-electron atoms, spin squeezing can be created in other atoms, such as, Sr atoms. For an example, see E. M. Bridge et al., "Rydberg spin-squeezing for a strontium optical lattice clock," 2014 European Frequency and Time Forum (EFTF), Neuchatel, 2014, pp. 439-442. doi: 10.1109/EFTF.2014.7331530.

Dear referees,

We hereby resubmit our manuscript NCOMMS-18-18113, entitled ‘Quantum-enhanced sensing using non-classical spin states of a highly magnetic atom’.

We thank the referees for their careful reading of the paper and for their positive reviews. We have addressed in details all their recommendations, please see our answers below.

Sincerely,

The authors.

Reviewer #1

We thank the referee 1 for his positive report.

Reviewer #2

- *The authors should clarify the confusion on the relation between and a spin-($N/2$) system and an ensemble of N spin-1/2 atoms. In lines 55-59, the authors mention a set of N spin-1/2 atoms behaves as an effective large spin of $J = N/2$, and the non-classical states can be prepared via interaction between atoms and cite Ref. [45]. In lines 63-68, the authors state that ‘this behavior results from the large size 2^N of the Hilbert space...’ Actually, for an ensemble of N identical bosons, which occupy two different internal spin states, they can be exactly mapped onto an effective large spin of $J = N/2$ and the corresponding Hilbert space is of $2J + 1 = N + 1$ dimensions but not 2^N dimensions. For an example, see PRL 97, 150402 (2006). The authors cite Ref. [45] for the preparation of non-classical states via inter-atom interactions. Actually, Ref. [45] only mentions the preparation of spin squeezing via time-evolution dynamics with fixed parameters. The authors should cite other related references on this topic, such as, the preparation of non-classical states via adiabatic evolutions (PRL 97, 150402 (2006); PRL 111, 180401 (2013); PRA 97, 032116 (2018)).*

An ensemble of N spins 1/2 evolving under a symmetric Hamiltonian can indeed be restricted to a Hilbert space of dimension $N + 1$. However, decoherence channels may couple it to non-symmetric states, then implying a Hilbert space of much larger dimension 2^N . To clarify this point, we explicitly state that the Hilbert space of dimension 2^N takes into account non-symmetric states (see last sentence of paragraph 3).

We agree that we should extend the reference list when introducing the generation of non-classical states via inter-atom interactions. We added to reference [45] ([43] in the new version) the references [34-37].

We also added the references to adiabatic evolutions in the conclusion, as a possible extension of our work (references [65-67]).

- *In lines 54-55, the authors mention that non-classical spin states has been created in ensemble of one-electron atoms. Actually, in addition to one-electron atoms, spin squeezing can be created in other atoms, such as, Sr atoms. For an example, see E. M. Bridge et al., ‘Rydberg spin-squeezing for a strontium optical lattice clock,’ 2014 European Frequency and Time Forum (EFTF), Neuchatel, 2014, pp. 439-442. doi: 10.1109/EFTF.2014.7331530.*

We modified the first sentence of paragraph 3, mentioning ensembles of one- and two-electron atoms. We would prefer keeping a single reference to the review [5], that lists in detail most of the previous works on the subject.